# Physicochemical Properties and Leaching Toxicity Assessment of Jarosite Residue

**Jun Peng** [1,2], **Hui Liu** [1,3], **Luhua He** [2], **Zhumei Sun** [1], **Yanmei Peng** [2], **Xiaofang Huang** [2] and **Xu Yan** [1,3,*]

1   School of Metallurgy and Environment, Central South University, Changsha 410083, China; pengjgy@163.com (J.P.); leolau@csu.edu.cn (H.L.); sunzhumei41@163.com (Z.S.)
2   Changsha Research Institute of Mining and Metallurgy Co., Ltd., Changsha 410012, China; hlh080358@163.com (L.H.); yanmei6345231@sina.com (Y.P.); xiaofangh@163.com (X.H.)
3   Chinese National Engineering Research Center for Control & Treatment of Heavy Metal Pollution, Changsha 410083, China
*   Correspondence: yanxu1202@csu.edu.cn; Tel.: +86-731-8883-0875

**Abstract:** The safe disposal of hazardous waste from zinc hydrometallurgy, such as jarosite residue, is crucial for the sustainable development of the industry. The chemical, structural and morphological properties of jarosite residue from zinc smelting were studied by a combination of various characterizations, and environmental stability was evaluated using the toxicity characteristic leaching procedure (TCLP), Chinese standard leaching tests (CSLT) and long-term leaching experiments (LTLE). Phase composition analysis revealed that zinc ferrite and sodium jarosite were the main phases present in the jarosite residue. TCLP and CSLT analyses indicated that the Zn and Pb contents exceeded their respective toxicity identification standards by more than 30 times and 8 times, respectively, exceeding the threshold values of the standard. The LTLE results demonstrated that Pb concentrations continued to exceed the standard limits, even after long contact times. This study has paramount significance in the prediction of jarosite residue stability and the evaluation of its potential for secondary environmental pollution.

**Keywords:** jarosite residue; mineralogical characteristics; hazardous; leaching toxicity evaluation; environmental stability





## 1. Introduction

Iron removal by the jarosite process is usually used in the zinc hydrometallurgy process, which has several advantages, such as simple operation, low production cost, good residue filtration performance, high efficiency, scalability and compatibility with existing processes. As a result, it has become the predominant iron removal process in zinc smelters both domestically and internationally [1,2]. Consequently, a large amount of jarosite residue is produced in zinc hydrometallurgy plants every year. According to statistics, a zinc hydrometallurgy plant with an annual output of 100,000 tons of electric zinc can annually generate 30,000–50,000 tons of jarosite residue [3]. In China alone, there are more than 30 million tons of jarosite residue stockpiled at present, and the amount is increasing year by year, which creates huge potential pollution for the environment [4]. Jarosite residue easily acts with Pb, Zn, Cd, Ag, In, As, etc. to form polymetallic coprecipitation, resulting in the loss of valuable metals, especially in the case of lead coprecipitation with jarosite [5]. When the coprecipitation of multiple metals with jarosite residue occurs in a heated environment or with a changed pH value, multiple metals such as Pb, As, and Cd are prone to decomposing and being released, causing serious environmental pollution and resource waste. Jarosite residue has been listed as hazardous waste in China and Europe [6]. Therefore, preventing and controlling pollution with the heavy metals in jarosite residue has become an important environmental protection task for the nonferrous metallurgical industry.

At present, two major measures are harmless treatment and resource utilization. Harmless treatment means that jarosite residue is used to replace part of the cement in the manufacturing of building materials or filling aggregates; thus some of the toxic and harmful elements can be fixed in building materials or filling materials [7,8]. However, due to the high sulfur content in jarosite residue, it is usually hard for the solidified body prepared by the general heavy metal solidification and stabilization technology to meet the requirements of product quality and performance, as well as environmental safety performance indicators at the same time. The produced building materials will still release heavy metals once they are damaged by weathering or acid rain erosion, thus bringing in harm to the ecological environment. As a result, it cannot be completely stable and harmless after treatment, and still potentially poses secondary pollution risks. In addition, a large number of metal elements with recycling value in the residue are fixed in building materials, resulting in the loss of valuable metal resources. There have been few research studies in this area, and most of the research at home and abroad usually focuses on the separation and recovery of valuable metals from jarosite residue. The high-temperature sintering method [9,10], reduction roasting-magnetic separation method [11,12], roasting-leaching method [13,14], solvent leaching method [15–17], microbial leaching method [18], self-sulfurization-flotation method [19], chlorination roasting method [20] and ionic liquid leaching method [21], among others, are usually adopted for the utilization of jarosite residue resources. The selection of the resource utilization method for jarosite residue mainly depends on the properties of the valuable components, phase composition and thermal decomposition characteristics of the jarosite residue. Therefore, a systematic study on the physicochemical properties of jarosite residue can not only provide a basis for the selection of resource utilization methods, but also reveal the key technical problems and breakthroughs of resource utilization from the source and microscopic point of view.

At present, the research on jarosite residue is mostly focused on the coprecipitation behavior of heavy metals in jarosite residue, while there has been research on the chemical forms and stability of heavy metals in jarosite residue during long-term stockpiling, and the effects of physicochemical properties such as the phase structure of jarosite residue on environmental characteristics, leaching toxicity and speciation distribution of heavy metals also have been ignored. In fact, the occurrence state and mineral phase structure of jarosite residue have greater influence on environmental pollution characteristics than the content of heavy metals. For example, the occurrence state of lead in jarosite residue will affect the pollution characteristics of the lead, which usually has low environmental activity if it is deposited in the lattice, but becomes more environmentally hazardous if it is present in an adsorbed state. Similarly, the varied phase structure of iron mineral will also affect the environmental activity of lead. The jarosite phase will be dissolved or recrystallized due to the change of environmental factors such as redox conditions, the solution pH value and microorganisms, etc. [22,23]. The change of its surface properties and crystal structure will release or re-fix the coprecipitated lead, thus affecting the migration law and environmental behavior of lead. Wu et al. [24] found that more Zn, Pb and Cu in the liquid phase were released after an addition of 10 mM $Fe^{2+}$ to the polymetallic jarosite at pH of 7, and the jarosite was transformed into goethite or magnetite. Smeaton et al. [25] inoculated an iron-reducing bacteria into the lead-containing jarosite; the results showed that the jarosite structure was destroyed after 336 h of incubation, and 12.4% of iron was released into the solution in the form of ferrous iron, while lead was transformed into cerussite and redistributed in the residue. Therefore, not only the content but also the occurrence state and phase of lead should be controlled in order to ensure the environmental stability of lead.

One of the main objectives of our study is to determine the characteristics of jarosite residue collected from zinc hydrometallurgy plants in China, including mineralogical characteristics, environmental migration and transformation behavior and pollution characteristics, which can enrich the database of basic physical and chemical characteristics of jarosite residue. It is supposed that the results we obtained can be used to predict the

stability of lead-containing jarosite residue and its secondary pollution for the environment, which is a must for sustainability.

## 2. Materials and Methods

### 2.1. Materials

The jarosite residue used in the study was obtained from a zinc smelter in the Guangxi Province of China, which was produced in the process of iron removal in zinc hydrometallurgy. The collected jarosite residue was dried to a constant weight at 105 °C, and then cooled to room temperature for later use. The appearance of the jarosite residue was grayish brown, as shown in Figure 1.

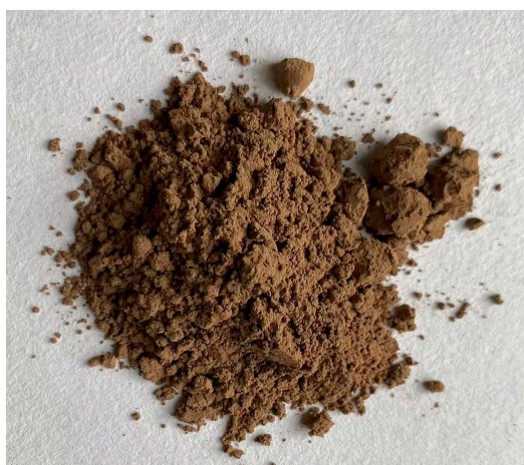

**Figure 1.** Appearance of the investigated jarosite residue.

### 2.2. Leaching Test

2.2.1. Toxicity Characteristic Leaching Procedure (TCLP) and China Standard Leaching Test (CSLT)

TCLP is a toxic leaching procedure designed by the Environmental Protection Agency (EPA) of the United States, commonly used to determine whether a solid waste is hazardous or not and also to evaluate its potential environmental risk. In addition, the China Identification Standard for Hazardous Wastes (HJ/T 299-2007; HJ/T 300-2007) [26,27], named CSLT-1 and CSLT-2, respectively, were compared with the TCLP method.

The sample treatment steps of the China standard leaching test (CSLT), including grinding, sieving, weighing, etc., are the same as those of the TCLP method. The main difference is in leaching agent and pH value. The comparison of several methods is listed in Table 1.

**Table 1.** Comparison of TCLP and CSLT Leaching Methods.

| Method | Leaching Reagent | Oscillation Time |
|---|---|---|
| TCLP | $CH_3COOH$ (acetic acid); pH = 2.88 ± 0.05 | At room temperature for (18 ± 2) h, 30 r/min |
| CSLT-1 (HJ/T 299-2007) | $H_2SO_4$:$HNO_3$ = 2:1; pH = 3.20 ± 0.05 | At room temperature for (18 ± 2) h, 30 r/min |
| CSLT-2 (HJ/T 300-2007) | $CH_3COOH$(glacial acetic acid); pH = 2.64 ± 0.05 | At room temperature For(18 ± 2) h, 30 r/min |

The CSLT-2 method was taken as an example to briefly describe the following steps. The sample was ground to a particle size of less than 9.5 mm and dried. Then, 1.00 g of the ground sample was weighed out and put into a 50 mL polyethylene centrifuge tube. A mixture of glacial acetic acid was added as the extraction agent, with the pH adjusted to 2.64 ± 0.05. The extraction bottles were sealed and placed in a standard tumbler, then

tumbled for (18 ± 2) h at a rotation speed of 30 r/min. After that, the samples were centrifuged for 20 min at 4000 r/min, and the supernatant was filtered through a 0.45 μm filter membrane. Then, the filtrate was analyzed to determine its heavy metal content, and the leaching toxicity of the sample was evaluated according to the identification standard of hazardous waste.

### 2.2.2. Long-Term Stability Test

A leaching toxicity test could evaluate only the existing properties of the jarosite residue, but the properties and components of jarosite residue may change after long-term stocking in the environment. Therefore, it is of great significance to evaluate the long-term stability of jarosite residue.

In order to simulate the long-term leaching effect of the jarosite residue under acid rain environment, a long-term leaching experiment (LTLE) was designed according to ANS (1986) standard [28]. A residual weighting of 30 g was used for each test, and the leaching agent (whereby deionized water was adjusted to a pH of 3.00 ± 0.05 with concentrated sulfuric acid, concentrated nitric acid and a concentrated acetic acid solution of mass ratio of 3:1:3) was used to simulate acid rain. The leaching process lasted for 20 d with a leaching speed of 1.25 mL/h. The leaching solution was collected every 24 h and filtered through a 0.45 μm filter membrane. Then, the content of heavy metals in the filtrate was analyzed to evaluate its long-term stability.

### *2.3. Analysis*
### 2.3.1. Determination of Elemental Composition

The chemical composition of the jarosite residue was detected firstly by X-ray fluorescence (XRF, Bruker, model S4-PIONER), and then the main elements such as iron, lead, zinc, copper, silver, sulfur, arsenic and cadmium were analyzed by atomic absorption spectroscopy (AAS) based on the XRF results.

The total content of the main elements could be obtained according to the following steps. The air-dried samples were oven-dried at 90 °C for 4 h, then crushed and sieved to powder with a particle size of 74 μm. After that, 0.2 g powder was accurately weighed out and placed in a 20 mL polyethylene tube with 10 mL of 6 mol/L HCl to stand at room temperature for 24 h for digestion. The solution was then diluted with deionized water in 50 mL volumetric flasks and analyzed by AAS.

### 2.3.2. Determination of PHASE Composition

The mineralogical composition of the samples was determined firstly by X-ray diffraction analysis (XRD, BRUKER, Karlsruhe, Germany, D8 ADVANCE) using Cu Kα radiation with steps of 0.02° at 10°/min in a 2θ range from 5° to 80° under operating conditions of 40 kV and 40 mA. Then, the phase composition including iron, lead, zinc and copper was investigated using chemical analysis based on the XRD results.

During the analysis, one main phase was dissolved in a pre-prepared specific solvent. The leaching solution was filtered with a vacuum filter after being dissolved completely and the filter residue was used for the analysis of another phase in the next stage. Meanwhile, the content of supernatant was analyzed by AAS. The processes of the chemical phase analysis of iron, lead, zinc and copper are shown in Figures 2–5 [29], respectively.

### 2.3.3. Other Analyses

The microstructure, surface morphology and special part chemical composition were observed by scanning electron microscope (SEM-EDS, JEOL, Tokyo, Japan, JSM-7900F). The grain morphology and mineralogical surface composition of the jarosite residue were examined by X-ray photoelectron spectroscopy (XPS, Thermo Fisher Scientific, Waltham, MA, USA, ESCALAB 250Xi) with an Al Kα X-ray source in a vacuum of $10^{-7}$ Pa. The molecular bonding structure and phase composition were studied using Fourier transform infrared spectroscopy (FTIR, Thermo Fisher Scientific, Waltham, MA, USA, IS50 FT-IR)

on KBr pellets (whereby the sample is ground into a fine powder and then mixed with KBr powder before the mixture is compressed under pressure to form a solid pellet) in the 400–4000 cm$^{-1}$ spectral range with 32 scans per spectrum at a resolution of 4 cm$^{-1}$. The particle size of the jarosite residue was analyzed by a laser particle size analyzer (Malvern, Panalytical, Malvern, UK, Mastersizer-2000). The specific surface area analysis measurement was determined by a fully automatic nitrogen adsorption specific surface instrument (BSD Instrument, Beijing, China, 3H-2000A).

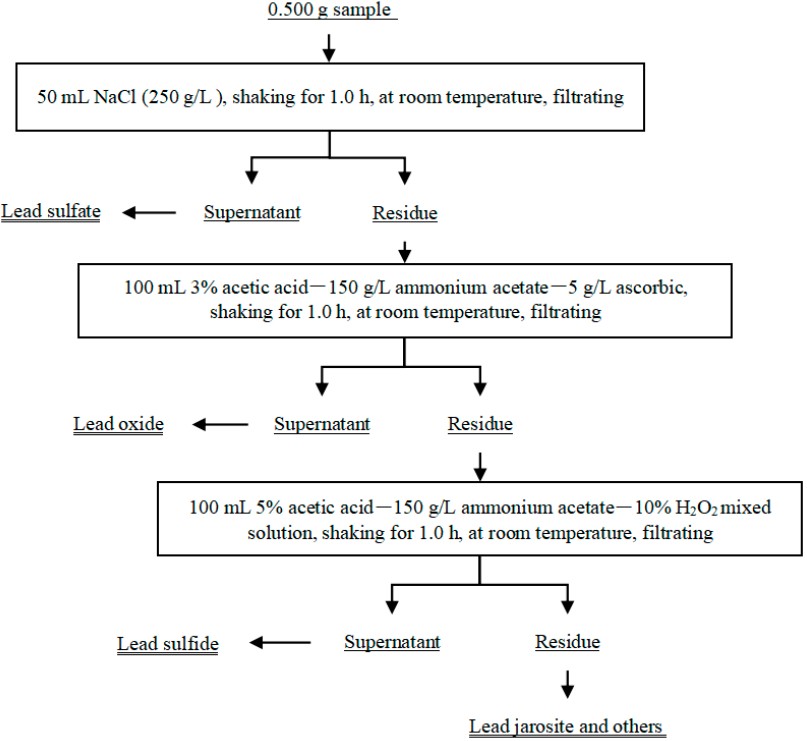

**Figure 2.** Methodology used for chemical phase analysis of lead in jarosite residue.

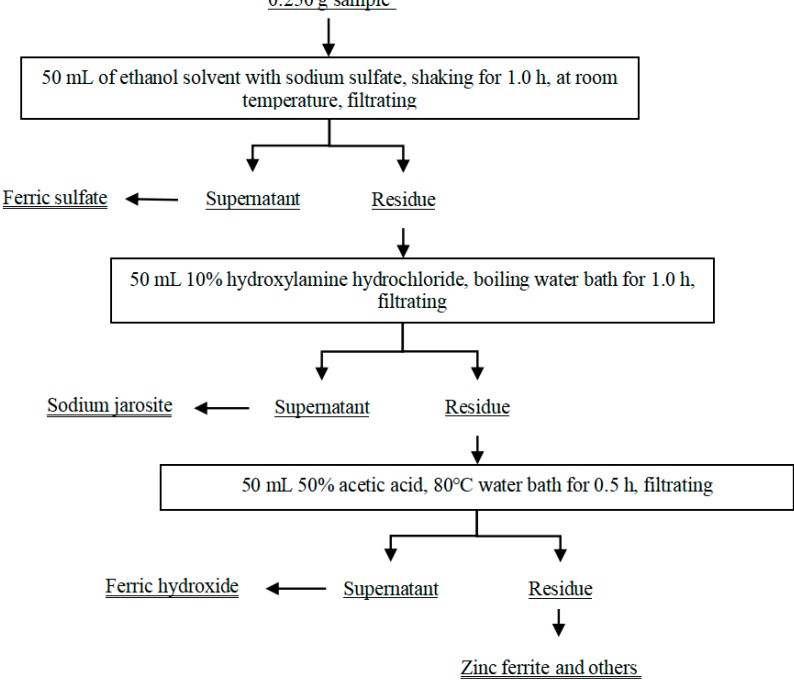

**Figure 3.** Methodology used for chemical phase analysis of iron in jarosite residue.

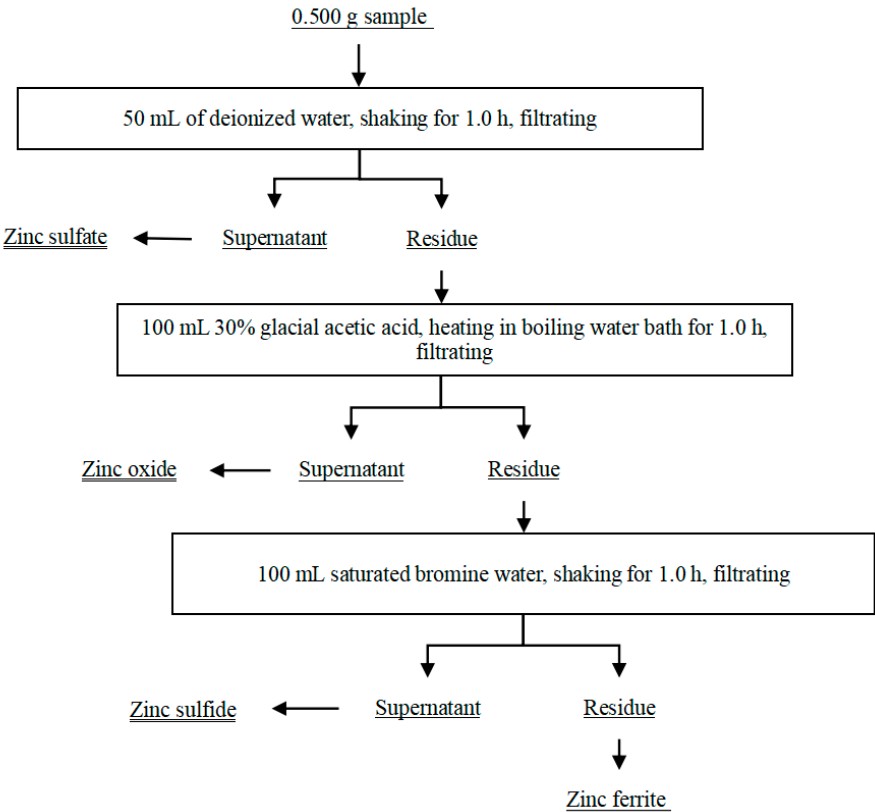

**Figure 4.** Methodology used for chemical phase analysis of zinc in jarosite residue.

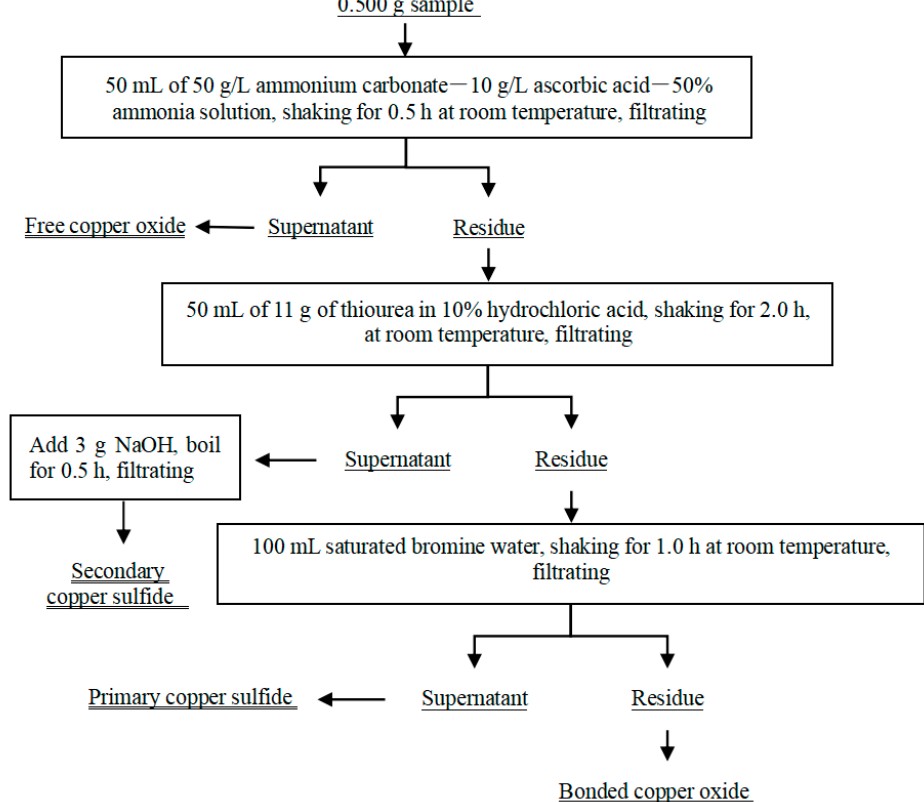

**Figure 5.** Methodology used for chemical phase analysis of copper in jarosite residue.

## 3. Results and Discussion

### 3.1. Chemical Composition

The elemental composition of the jarosite residue sample was analyzed by XRF and the chemical analysis results are given in Tables 2 and 3, respectively. The data from the ICP-AES are similar to those from the XRF. The main elements in the jarosite residue were Fe, O, S, Zn, Pb and Na, which accounted for more than 95% of the total mass of the jarosite residue. In addition, the jarosite residue also contained a small amount of Mn, Cu, As, Sn, Cd, Ag, etc., among which Zn, As, Pb, Cu, Cd are toxic and harmful elements and should not be released into the environment.

**Table 2.** Chemical composition of jarosite residue obtained by XRF (mass fraction, %).

| O | Fe | S | Zn | Na | Pb | Mn | Cu | Al | Sn |
|---|---|---|---|---|---|---|---|---|---|
| 39.31 | 32.92 | 11.33 | 9.11 | 2.61 | 2.99 | 0.60 | 0.36 | 0.18 | 0.18 |
| K | Si | P | As | Ba | Ti | Cd | Co | Bi | Cr |
| 0.16 | 0.06 | 0.02 | 0.03 | 0.03 | 0.02 | 0.03 | 0.02 | 0.02 | 0.01 |

**Table 3.** Chemical composition of jarosite residue obtained by chemical analysis (mass fraction, %).

| Fe | Zn | Pb | Cu | Ag | Cd | S | C | As | Sn |
|---|---|---|---|---|---|---|---|---|---|
| 33.99 | 8.08 | 2.81 | 0.35 | 0.017 | 0.022 | 9.09 | 0.047 | 0.027 | 0.164 |

### 3.2. Phase Analysis

Figure 6 shows the XRD pattern of the jarosite residue. It was found that sodium jarosite ($NaFe_3(SO_4)_2(OH)_6$, PDF #36-4025), zinc ferrite ($ZnFe_2O_4$, PDF #22-1012) and lead sulfate ($PbSO_4$, PDF #36-1461) were the main phases present in the jarosite residue samples. Among the high peaks, most of the Fe existed in the sodium jarosite, Zn existed in the form of zinc ferrite and Pb existed in the form of lead sulfate. Zinc ferrite is usually formed in the process of the oxidation roasting of zinc sulfide concentrate, and zinc calcine containing zinc ferrite is usually used as neutralizer during the jarosite process, so that the zinc ferrite enters into the jarosite residue as it is difficult for it to be dissolved. However, the pollution-free jarosite process can avoid the zinc ferrite entering into the jarosite residue without using zinc calcine as a neutralizer. Therefore, not all jarosite residue contains a zinc ferrite phase. Lead oxide in zinc calcine reacts with sulfuric acid in the jarosite process to form insoluble lead sulfate, and the lead jarosite phase may also be formed according to Ref. [5]. Thus, in order to improve phase identification in jarosite residue samples, it is necessary to use other techniques, such as chemical phase analysis, SEM-EDS and XPS.

The analysis results of the chemical phase composition of iron, lead, zinc and copper in the jarosite residue are shown in Tables 4–7 respectively. It is seen from Table 3 that 71.82% of iron exists in the form of sodium jarosite, while the zinc ferrite phase and others account for 23.77%. The analysis results in Table 4 show that 77.22% of lead exists in the form of lead jarosite and 17.79% in the form of lead sulfate phase, while 2.56% and 2.42% exist in the form of lead oxide and lead sulfide, respectively. The lead in the chemical phase analysis results is different from that of the XRD, mainly because the occurrence state of lead is complex, and XRD cannot accurately identify a low-grade lead phase. Based on the results of both analyses, it is found that lead mainly exists in the form of lead jarosite and lead sulfate. According to the analysis results in Table 5, 81.31% of zinc exists in the form of zinc ferrite, while 16.83%, 0.37% and 1.49% exist in the form of the zinc oxide phase, zinc sulfate and zinc sulfide, respectively. Table 5 shows that up to 96.00% of copper exists as a bound copper oxide phase, while only 4.00% exists in the other phases, including free copper oxide, secondary copper sulfide and primary copper sulfide.

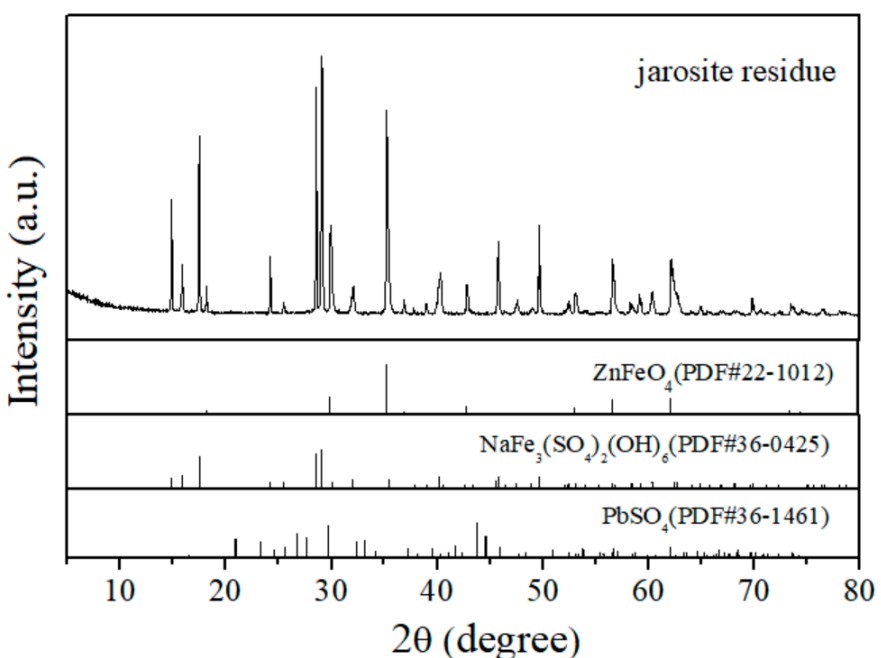

**Figure 6.** XRD patterns of jarosite residue.

**Table 4.** Phase composition of iron in jarosite residue obtained by chemical analysis.

| Phase | w (Fe), % | Phase Percentage, % |
|---|---|---|
| Ferric sulfate | 0.05 | 0.15 |
| Sodium jarosite | 24.41 | 71.82 |
| Ferric hydroxide | 1.45 | 4.27 |
| Zinc ferrite and others | 8.08 | 23.77 |
| Total | 33.99 | 100 |

**Table 5.** Phase composition of lead in jarosite residue obtained by chemical analysis.

| Phase | w (Pb), % | Phase Percentage, % |
|---|---|---|
| Lead sulfate | 0.50 | 17.79 |
| Lead oxide | 0.072 | 2.56 |
| Lead sulfide | 0.068 | 2.42 |
| Lead jarosite and others | 2.17 | 77.22 |
| Total | 2.81 | 100 |

**Table 6.** Phase composition of zinc in jarosite residue obtained by chemical analysis.

| Phase | w (Zn), % | Phase Percentage, % |
|---|---|---|
| Zinc sulfate | 0.03 | 0.37 |
| Zinc oxide | 1.36 | 16.83 |
| Zinc sulfide | 0.12 | 1.49 |
| Zinc ferrite | 6.57 | 81.31 |
| Total | 8.08 | 100 |

**Table 7.** Phase composition of copper in jarosite residue obtained by chemical analysis.

| Phase | w (Cu), % | Phase Percentage, % |
|---|---|---|
| Free copper oxide | 0.003 | 0.86 |
| Secondary copper sulfide | 0.010 | 2.86 |
| Primary copper sulfide | 0.001 | 0.29 |
| Bonded copper oxide | 0.336 | 96.00 |
| Total | 8.08 | 100 |

*3.3. Structural Feature*

Microstructural analysis of the jarosite residue was carried out by using a scanning electron microscope (SEM), and the results are shown in Figures 7 and 8. Based on these microphotographs and EDS, it can be seen that the phase composition and the particle size of jarosite residue are not homogeneous. Figure 7 shows that there are two kinds of morphology for the jarosite residue: one is rhombohedral, octahedral or a flake with a grain size of 1 μm~5 μm, which is consistent with Ref. [24]; the other comprises smooth ellipsoid or irregular small particles with a particle size of 0.1 μm~5 μm, which is mainly attached to the surface of rhombic large particles. It can be seen that the rhombohedral crystals grow in overlapping agglomeration, and the smooth ellipsoidal small particles are wrapped or sandwiched on the surface in region B. Figure 8 shows a typical phase distribution of jarosite. The two types of jarosite slag morphology exhibit distinct colors. The rhombohedral crystals appear dark, while the ellipsoidal particles exhibit a bright white coloration. The bright white points, identified as points 2 and 5, share a similar composition primarily composed of O, Fe, and Zn. This composition strongly suggests that these bright white points could be zinc ferrite. Importantly, it is worth noting that zinc ferrite does not contain lead, as indicated by its composition. On the other hand, the dark spots identified as points 1, 4, 6, and 7 exhibit similar compositions. They primarily consist of O, S, Fe, Na, and Pb. This composition strongly suggests that these dark spots are likely composed of jarosite.

In order to identify the internal morphology and element distribution of the jarosite residue, the samples were cured with triethanolamine-epoxy resin and analyzed by SEM-EDS after slicing, grinding and polishing, and the results are shown in Figure 9. it was observed that the bright white points (1, 3, 7, 8, and 11) exhibited a striking similarity in composition. These points contained O, Fe, and Zn elements, but interestingly, lacked the presence of Pb. This compositional pattern strongly indicated the presence of zinc ferrite within these points. Notably, the zinc ferrite phase exhibited a decentralized structure, showcasing superfine particles spanning a diameter range of 1μm to 10μm. Contrastingly, the gray points (2, 4, 5, 6, 9, and 10) were identified as sodium jarosite and lead jarosite based on their composition, which encompassed O, S, Fe, Na, and Pb elements. It is noteworthy that the detection of lead jarosite through XRD analysis was challenging due to its amorphous state and its ultrafine particle size.

A comprehensive analysis of the spatial distribution of the key elements in the jarosite residue was performed using secondary electron scanning and X-ray mapping techniques. The result, depicted in Figure 10, indicates that the distribution regions of the Fe, O, Zn and Na elements partly overlap with each other, and Fe distribution is highly concentrated in the regions where the O, Zn, and O, Na elements are present, which may indicate that Fe is mainly present in the form of zinc ferrite or jarosite. The distribution regions of the Pb, S, O and Na elements overlap, and Pb is mainly distributed in the S, O and Na, O elements, which indicates that Pb may mainly exist in the form of lead sulfate and lead jarosite. In addition, copper and manganese are dispersed at a low concentration.

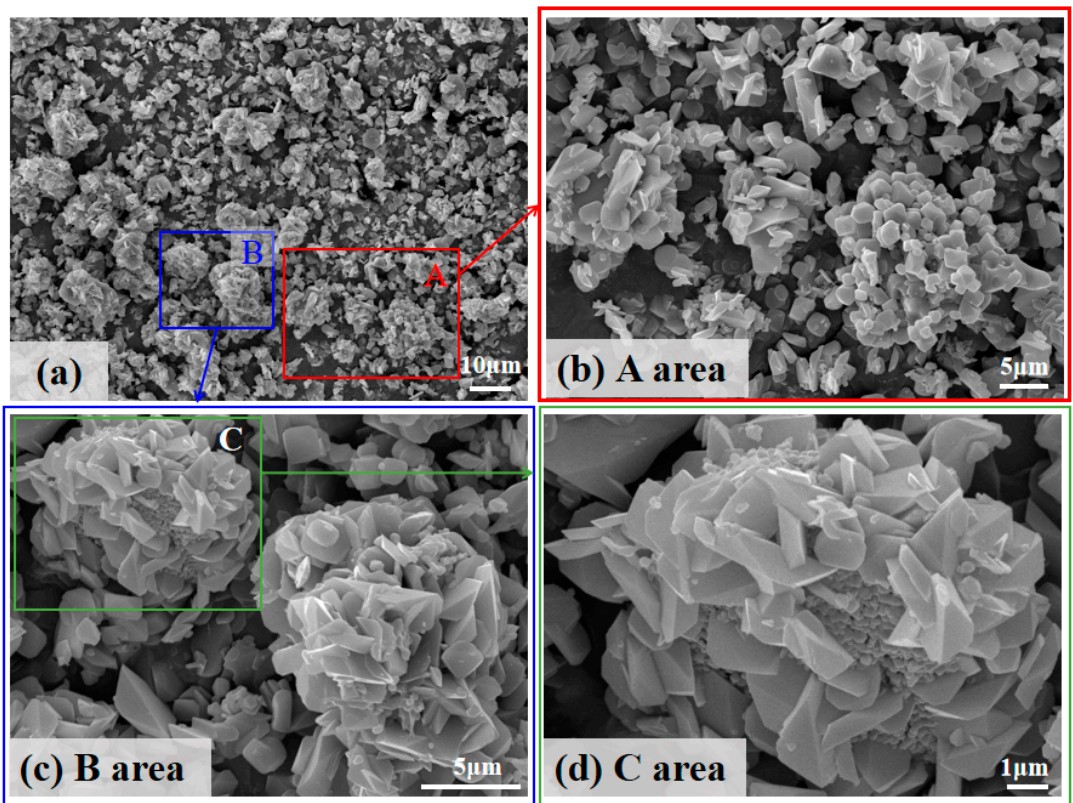

**Figure 7.** SEM images of jarosite residue. (**a**) Mag. 1000×; (**b**) Area A, Mag. 3000×; (**c**) Area B, Mag. 5000×; (**d**) Area C, Mag. 10,000×.

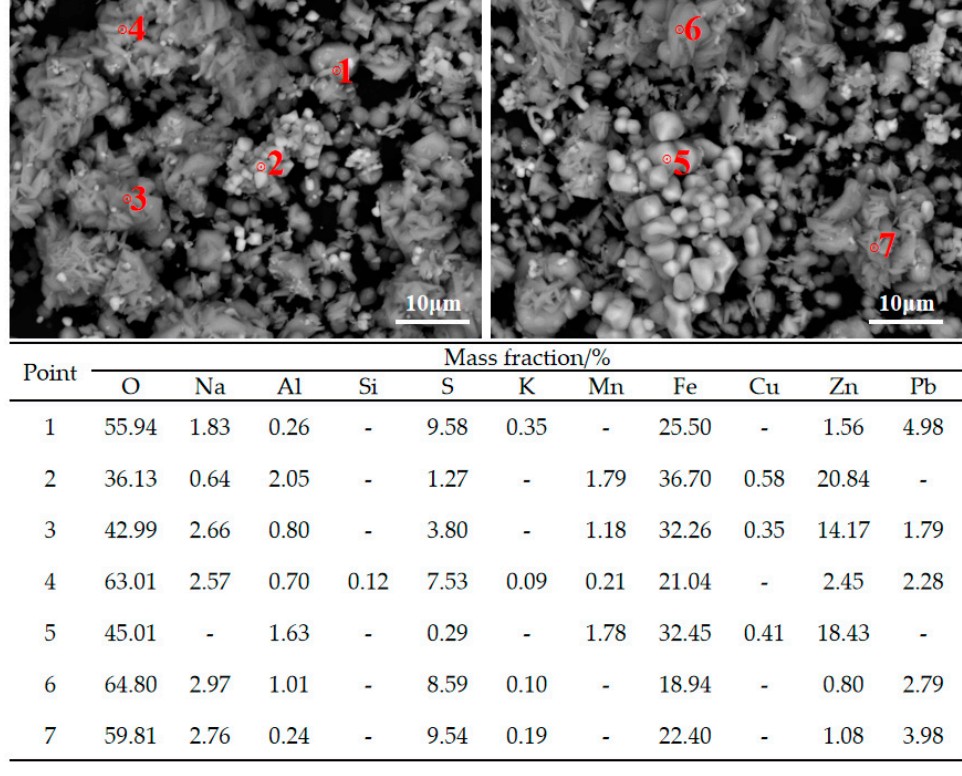

| Point | Mass fraction/% | | | | | | | | | | |
|-------|------|------|------|------|------|------|------|------|------|------|------|
|       | O | Na | Al | Si | S | K | Mn | Fe | Cu | Zn | Pb |
| 1 | 55.94 | 1.83 | 0.26 | - | 9.58 | 0.35 | - | 25.50 | - | 1.56 | 4.98 |
| 2 | 36.13 | 0.64 | 2.05 | - | 1.27 | - | 1.79 | 36.70 | 0.58 | 20.84 | - |
| 3 | 42.99 | 2.66 | 0.80 | - | 3.80 | - | 1.18 | 32.26 | 0.35 | 14.17 | 1.79 |
| 4 | 63.01 | 2.57 | 0.70 | 0.12 | 7.53 | 0.09 | 0.21 | 21.04 | - | 2.45 | 2.28 |
| 5 | 45.01 | - | 1.63 | - | 0.29 | - | 1.78 | 32.45 | 0.41 | 18.43 | - |
| 6 | 64.80 | 2.97 | 1.01 | - | 8.59 | 0.10 | - | 18.94 | - | 0.80 | 2.79 |
| 7 | 59.81 | 2.76 | 0.24 | - | 9.54 | 0.19 | - | 22.40 | - | 1.08 | 3.98 |

**Figure 8.** BSE image of zinc leaching residue.

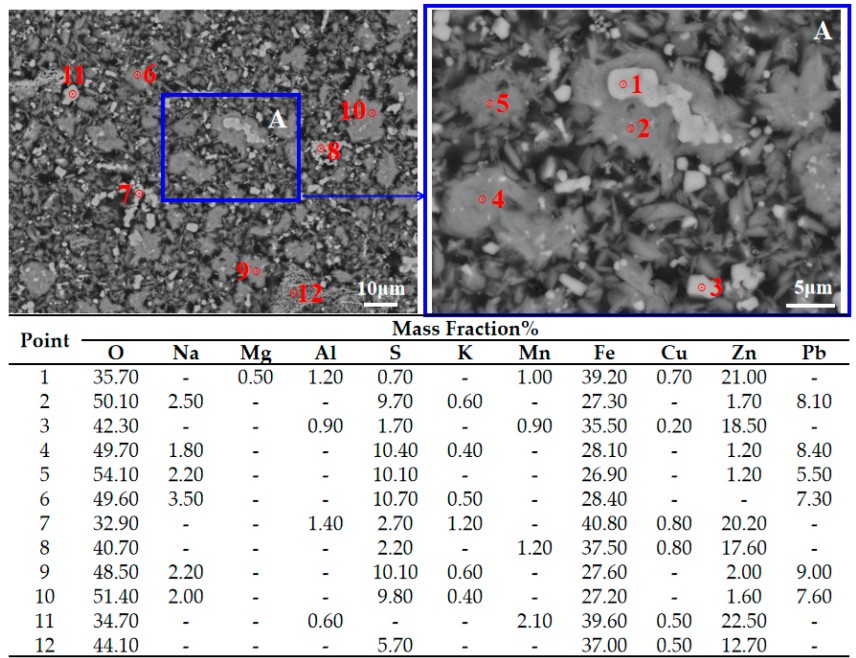

| Point | Mass Fraction% | | | | | | | | | | |
|---|---|---|---|---|---|---|---|---|---|---|---|
| | O | Na | Mg | Al | S | K | Mn | Fe | Cu | Zn | Pb |
| 1 | 35.70 | - | 0.50 | 1.20 | 0.70 | - | 1.00 | 39.20 | 0.70 | 21.00 | - |
| 2 | 50.10 | 2.50 | - | - | 9.70 | 0.60 | - | 27.30 | - | 1.70 | 8.10 |
| 3 | 42.30 | - | - | 0.90 | 1.70 | - | 0.90 | 35.50 | 0.20 | 18.50 | - |
| 4 | 49.70 | 1.80 | - | - | 10.40 | 0.40 | - | 28.10 | - | 1.20 | 8.40 |
| 5 | 54.10 | 2.20 | - | - | 10.10 | - | - | 26.90 | - | 1.20 | 5.50 |
| 6 | 49.60 | 3.50 | - | - | 10.70 | 0.50 | - | 28.40 | - | - | 7.30 |
| 7 | 32.90 | - | - | 1.40 | 2.70 | 1.20 | - | 40.80 | 0.80 | 20.20 | - |
| 8 | 40.70 | - | - | - | 2.20 | - | 1.20 | 37.50 | 0.80 | 17.60 | - |
| 9 | 48.50 | 2.20 | - | - | 10.10 | 0.60 | - | 27.60 | - | 2.00 | 9.00 |
| 10 | 51.40 | 2.00 | - | - | 9.80 | 0.40 | - | 27.20 | - | 1.60 | 7.60 |
| 11 | 34.70 | - | - | 0.60 | - | - | 2.10 | 39.60 | 0.50 | 22.50 | - |
| 12 | 44.10 | - | - | - | 5.70 | - | - | 37.00 | 0.50 | 12.70 | - |

**Figure 9.** SEM-EDS images of jarosite residue after slicing.

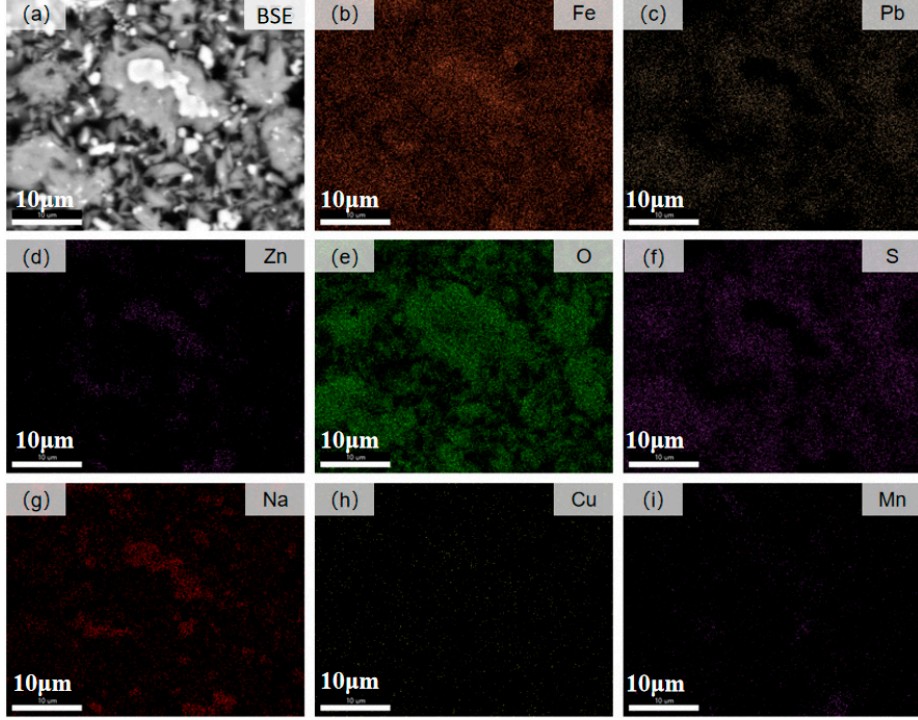

**Figure 10.** BSE image (**a**) and elements (**b–i**) distribution of jarosite residue.

### 3.4. Molecular Bonding Structure

The infrared spectrum of the jarosite residue is given in Figure 11, and data analysis was performed according to the results and relevant manuals [30–32]. It is shown that the stretching vibration characteristic peak of hydrogen bonded water (H–O) is at the spectrum of 3361.12 cm$^{-1}$, while the vibration of HOH deformation produced by water molecule deformation is at 1635.37cm$^{-1}$, and the bending vibration characteristic peaks of O–H deformation are at 1022.10 cm$^{-1}$ and 1009.26 cm$^{-1}$. The characteristic peaks of 476.20 cm$^{-1}$ and 507.14 cm$^{-1}$ correspond to the Fe–O bonds in octahedral positions, and the

characteristic peak of 507.14 cm$^{-1}$ may also be the stretching vibrations of Zn–O bonds in tetrahedral positions, which indicates the existence of simple oxide and spines, such as zinc ferrite and zinc oxide. The strong bands located at 1189.38 cm$^{-1}$ and 1093.64 cm$^{-1}$ are the stretching vibration characteristic peaks of v$_3$(SO$_4$), and the shoulder peak at 629.60 cm$^{-1}$ is the stretching vibration characteristic peak of v$_4$(SO$_4$), which is consistent with the corresponding v$_3$(SO$_4$) (1190 cm$^{-1}$ and 1100 cm$^{-1}$) and v$_4$(SO$_4$) (628 cm$^{-1}$) for jarosite.

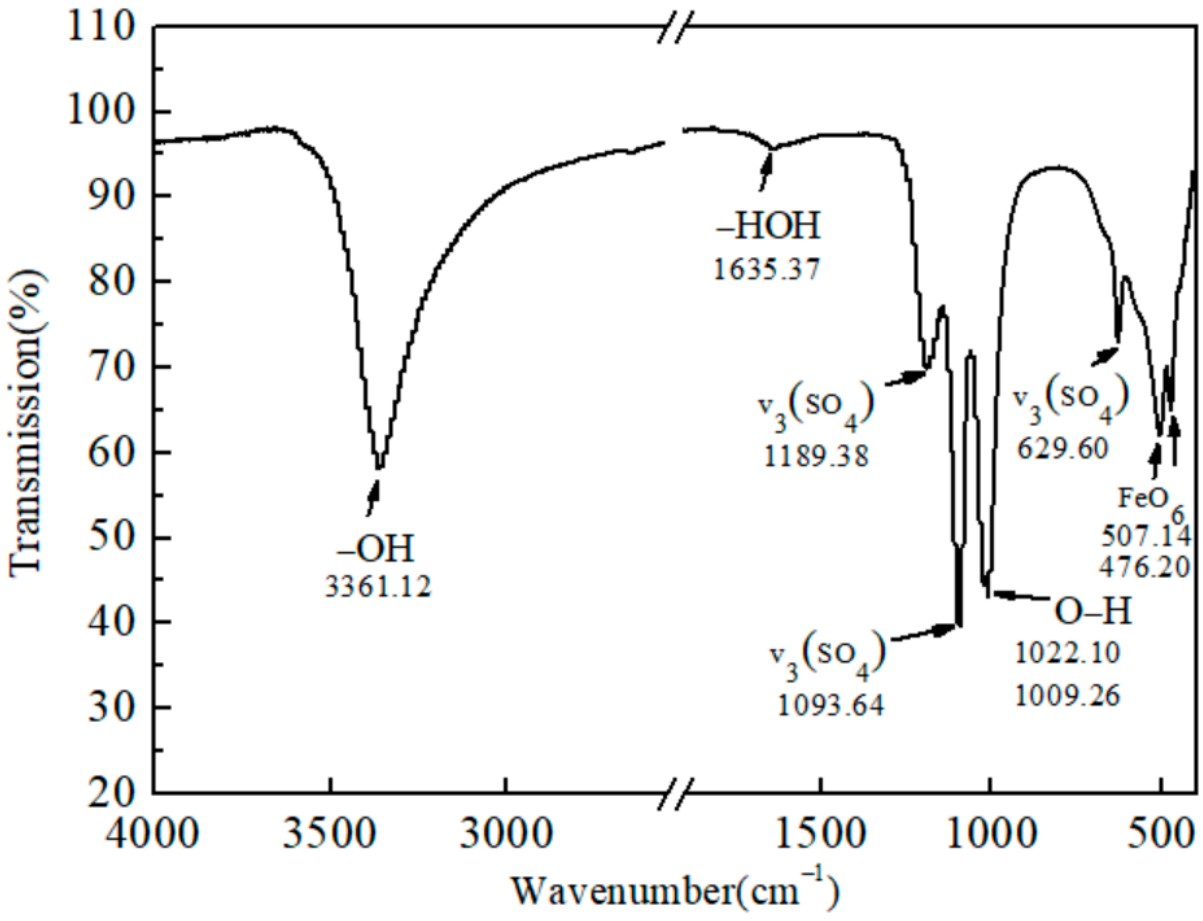

**Figure 11.** FTIR spectra of jarosite residue.

*3.5. Surface Performance*

It is very important to investigate the surface performance of jarosite residue due to the fact that most solid waste treatments are initially controlled by surface chemical reactions. The spectra of Fe2p, O1s, S2p, Zn2p, Pb4f and full spectrum peaks are displayed in Figure 12, respectively. XPS analysis was conducted to command elemental composition and oxidation states on the surface in line with relevant references [33,34].

The iron spectrum in Figure 12A with the binding energies of Fe 2p3/2 and Fe 2p1/2 are measured as 711.9 eV and 726.1 eV, respectively, the presence of which indicates the presence of zinc ferrite and jarosite [35]. The electron binding energy of Zn 2p1/2, shown in Figure 12D, is 1044.8 eV, which may correspond to the characteristic peak of ZnFe$_2$O$_4$. The fitting zinc spectrum of Zn 2p3/2 indicates the presence of ZnSO$_4$ and ZnO with binding energies of 1023.6 eV and 1021.7 eV, respectively. The Pb 4f7/2 peak separations in Figure 12E show the electron binding energy of 138.8 eV, 137.4 eV and 139.4 eV, which means the presence of PbO, PbO$_2$ and PbSO$_4$, respectively. These peak separations indicate that the particle surface of the jarosite residue is wrapped by ZnFe2O4, PbSO$_4$, NaFe$_3$(SO$_4$)$_2$(OH)$_6$, PbO and ZnO; thus it is difficult to separate valuable metals and recover them from jarosite residue due to the complex surface properties and phase composition.

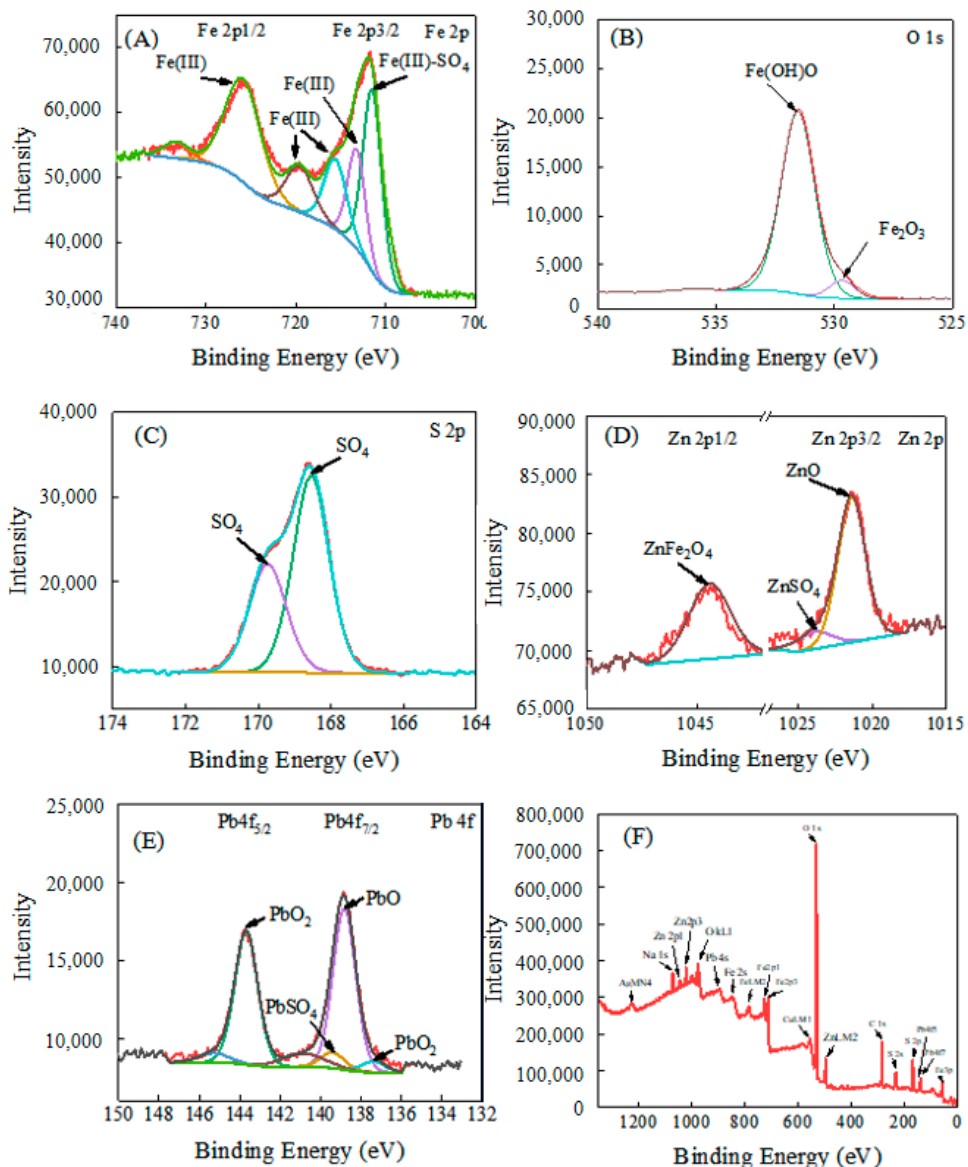

**Figure 12.** XPS spectra of jarosite residue. (**A**) Fe2p; (**B**) O1s; (**C**) S2p; (**D**) Zn2p; (**E**) Pb4f; (**F**) full spectrum.

### 3.6. Particle Size and Surface Area Features

Figure 13 shows the particle size distribution within the range from 0 to 100 μm. The size of the jarosite residue particles is distributed in two concentrated areas of approximately 2.73 and 13.76 μm. It is shown that most particles are small, with size d (0.5) of 3.883 μm and d (0.9) of 18.478 μm, which it is difficult to process using mineral processing, and which must be treated by metallurgical processing. In addition, the volume average particle size D (4,3) of 7.407 μm and the surface area average particle size D (3,2) of 2.326 μm indicate that the particles of the jarosite residue are very fine, and can be processed without additional grinding.

BET specific surface area analysis was performed for the jarosite residue and the results are shown in Table 8. The results show that the specific surface area of the jarosite residue calculated by different methods is between 1.9 and 2.2 m$^2$/g, and almost no micropore is less than 2 nm.

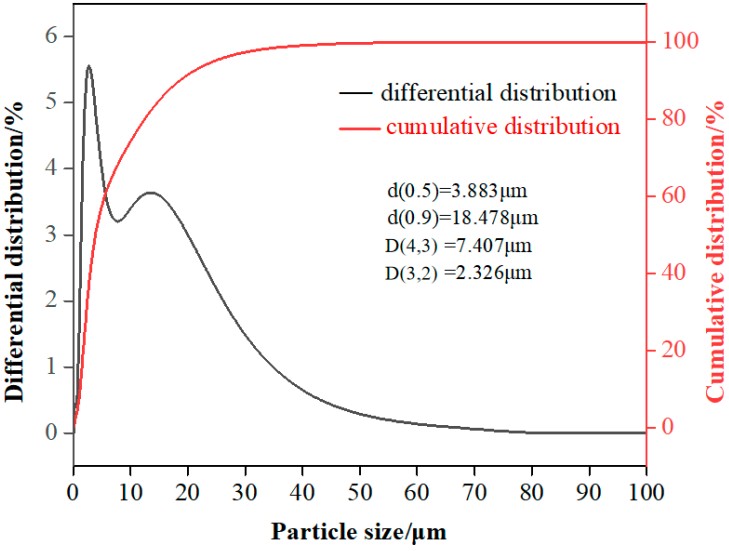

**Figure 13.** Particle size distribution of jarosite residue.

**Table 8.** BET specific surface area of jarosite residue.

| Surface Area (m$^2$/g) | | | Pore Volume (cm$^3$/g) | |
|---|---|---|---|---|
| Single point BET specific surface area (P/P$_0$0.2) | Multipoint BET specific surface area | Internal surface area of pores by T-diagram method | Outside surface area of pores by T-diagram method | Total volume of wells (<2 nm) by T-plot |
| 2.170 | 2.194 | 0.265 | 1.929 | 0.00012 |

### 3.7. Environmental Stability

The leaching toxicity results of heavy metals in the jarosite residue are shown in Table 9. The results show that both Zn and Pb released from the jarosite have concentrations higher than the regulatory threshold values of the US Environmental Protection Agency (USEPA) and the Chinese regulatory GB 5085.3-2007 (CRGB). The lead concentration in the jarosite is more than 30 times higher than the USEPA and CRGB standards. The solubility of Pb in the jarosite residue identified by the CSLT-1 method is lower than that of the other two methods, due to the formation of lead sulfate precipitation. The concentration of Zn released from the jarosite is basically the same using the three methods, which is eight times higher than the limit of 5.0 mg/L in USEPA and CRGB regulatory.

**Table 9.** Leaching concentrations of jarosite residue (mg/L).

| Element | Regulatory Threshold (China) | Regulatory Threshold (USEPA) | Leaching Concentrations | | |
|---|---|---|---|---|---|
| | | | TCLP Method | CSLT-1 Method | CSLT-2 Method |
| Zn | ≤100 | N [1] | 832.24 | 872.42 | 857.11 |
| Pb | ≤5 | ≤5 | 176.77 | 26.18 | 185.82 |
| Cd | ≤1 | ≤1 | 0.76 | 0.94 | 0.85 |
| Be | ≤0.02 | N [1] | - | - | - |
| Cu | ≤100 | N [1] | 90.62 | 92.65 | 96.89 |
| Ag | ≤5 | ≤5 | 2.28 | 3.83 | 4.77 |
| Hg | ≤0.1 | ≤0.2 | - | - | - |
| Cr | ≤15 | ≤5 | 5.87 | 6.92 | 6.63 |
| As | ≤5 | ≤5 | 2.87 | 4.24 | 4.72 |

[1] Not limited in the standard; - Not detected.

Long-term stacking of jarosite residue in the environment may pose environmental pollution risks due to heavy metal redissolution. Therefore, the results of the long-term leaching experiment (LTLE) are shown in Figure 14, which demonstrates that the initial stages of leaching revealed elevated levels of Zn, Cu and Pb, particularly for Zn, which can be attributed to the leaching of zinc oxide and zinc sulfate. Over the course of continuous leaching, the Zn concentrations rapidly decreased within 72 h, while the concentration of Zn in the leaching solution remained relatively stable after 72 h. However, the jarosite residue exhibited a residual Pb concentration (>5 mg/L) after the leaching test, suggesting that it may have the potential to continually pollute the environment.

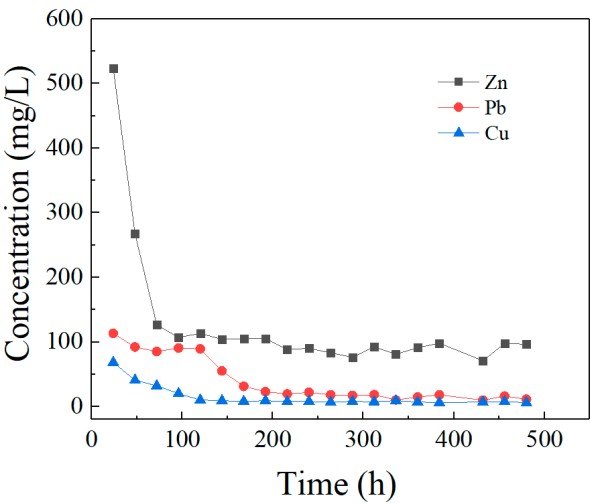

**Figure 14.** Variation of leaching concentration of heavy metals with time in LTLE.

## 4. Conclusions

(1) The mineralogical analysis indicates that the jarosite residue is mainly composed of zinc ferrite and sodium jarosite. The major phase constituents of the zinc, lead, copper and iron in the jarosite are zinc ferrite, lead jarosite, combined copper oxide and sodium jarosite, accounting for 81.31%, 77.22%, 96.00% and 71.82% of the total amount, respectively.

(2) The leaching toxicity study shows that the contents of lead and zinc in the leaching solution are 30 and 8 times higher than the regulatory limit using both the TCLP and CSLT assessment methods for contact over a short time, which could be classified as hazardous solid waste by the EPA or Chinese government. The long-term stability experiments show that the lead concentration exceeds the standard significantly, which indicates that the jarosite residue is a hazardous waste.

(3) The use of various methods to characterize the jarosite residue increases the reliability of the results. Meanwhile, these results also provide more comprehensive mineralogical data for jarosite residue, which can enhance the possibility of harmless treatment and resource utilization of jarosite residue in the zinc hydrometallurgy industry.

**Author Contributions:** J.P. and X.Y. conceived and designed the experiments; J.P., X.Y. and Z.S. analyzed the data; J.P. wrote and edited the paper; X.Y. and H.L. revised the work critically for important intellectual content; X.H. and Y.P. polished the paper; conceptualization: J.P. and L.H.; methodology: J.P., L.H. and Y.P.; software: J.P. and Z.S.; validation: J.P., L.H. and Y.P.; All authors have read and agreed to the published version of the manuscript.

**Funding:** This work was supported by the Major Program Natural Science Foundation of Hunan Province of China (No. 2021JC0001), National Natural Science Foundation of China (No. 22276218) and National Key R&D Program of China (No. 2021YFC2902804 and No. 2022YFC2904603).

**Data Availability Statement:** The original data used in this study cannot be made publicly available due to privacy and ethical considerations. Additionally, some of the funding projects related to this

study are still ongoing and have not yet been concluded. Therefore, the data cannot be released at this time. We assure readers that our conclusions and findings have been reached through rigorous analysis of the available data and robust methodologies. We will reevaluate the possibility of sharing the data once all relevant funding projects have been concluded and all privacy and ethical considerations have been addressed.

**Acknowledgments:** We gratefully acknowledge the many important contributions from the researchers of all the reports cited in our paper.

**Conflicts of Interest:** The authors declare no conflict of interest.

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
