# Peer review of "Physicochemical Properties and Leaching Toxicity Assessment of Jarosite Residue"

_sustainability, doi:10.3390/su15129472_

Round 1

Reviewer 1 Report

Lines 2-3, Title: It is ok.

Lines 14-26, Abstract: It is acceptable but it needs minor modifications especially in the last three lines, which concerns with the aim of study.

Line 27: You need to enhance/extend your keywords with one or two more, e.g. hazardous, Zn metallurgy.

Line 110, Fig. 1: Caption needs little modification.

Line 122, Table 1: For all chemical compounds in the entire text, you need to subscript when needed.

Line 148, Analysis: When you mention a technique for the first time, you need to give the full name first then abbreviate between two brackets, e.g. here X-ray fluorescence (XRF). You need to follow the same for the rest of techniques used in your study. You need to follow the correct way you reported XRD in lines 158-159, and SEM in line 184.

Lines 148-156, and all over the manuscript, please always consider space between digits and units of temperature, time, weight, volume, …etc.

Lines 171-180: The flow-sheets for methodologies shown in Figs 2 to 5 are good and only the issue of spacing between digits of weight and the unit exists.

Line 188: It is recommended to give a line or two about the preparation of the powdered pellets for the FTIR runs using KBr as a binding material.

Line 202, Table 2: It is much better if you present the output of the XRF analysis in the form of oxide wt.%, e.g. xx.x wt% CaO. The same is applicable for Table 3, and you can also use ppm for minor/trace elements that are in lower concentrations in the jarosite residue except for Zn.

Line 287: Starting from this line, there is an issue concerning editing, which is different spacing. Please be uniform and follow the instructions for authors.

Generally, data interpretations and conclusions are good.

It is acceptable English and needs fine polishing only.

Reviewer 2 Report

Physicochemical Properties and Leaching Toxicity Assessment of Jarosite Residue

This article focuses on the physicochemical properties and leaching toxicity assessment of jarosite residue, a byproduct of metal extraction processes. The authors discuss the chemical forms and stability of heavy metals in jarosite residue during long-term stockpiling, as well as the effects of physicochemical properties such as phase structure on environmental characteristics, leaching toxicity, and speciation distribution of heavy metals. The article also highlights the potential environmental impacts of improper disposal of jarosite residue and suggests effective methods for its safe disposal to minimize its environmental impact.

1.       The first paragraph provides a good introduction to the topic, highlighting the advantages of the jarosite process for iron removal in zinc hydrometallurgy. However, it would be helpful to mention specific advantages such as high efficiency, scalability, or compatibility with existing processes.

2.       Line 35: When providing statistics, it would be more informative to include the source or reference for the information regarding the annual output of electric zinc and the corresponding jarosite residue generation. This would enhance the credibility of the statement.

3.       It is mentioned that a large number of metal elements with recycling value are fixed in building materials during the harmless treatment process, resulting in the loss of valuable metal resources. To reinforce this point, provide examples or data to illustrate the extent of metal loss and its impact on resource sustainability.

4.       It would be helpful to briefly explain how the properties of valuable components, phase composition, and thermal decomposition characteristics of jarosite residue influence the selection of resource utilization methods. This would provide more insight into the decision-making process and the importance of understanding the physicochemical properties of jarosite residue.

5.       In the last paragraph mentions that the research aims to enrich the database of basic physical and chemical characteristics of jarosite residue and predict its stability and potential secondary pollution to the environment. Consider briefly explaining the potential applications or implications of this enriched database in terms of environmental management or remediation strategies.

6.       Please write the chemical formulas of the chemicals in good form throughout the manuscript.ie Table 1.

7.       Figure 14: please put the units of the axes between brackets.

8.       Begin names in tables with capital letters.

9.       In addition to the mentioned conclusions, it would be valuable to include some perspectives to guide future research and potential applications:

·         An important perspective would be to explore innovative methods for the treatment and valorization of jarosite residue. This could involve approaches such as utilizing bioleaching techniques or advanced extractive metallurgy to efficiently recover precious metals from the residue.

·         Another promising perspective would be to assess the possibilities of reusing or recycling jarosite residue in other industries or sectors. For instance, investigating its potential use as a construction material or as a source of alternative energy.

·         It would also be worthwhile to explore the economic and environmental aspects associated with the management of jarosite residue. A thorough cost-benefit analysis would provide a better understanding of the potential advantages of different treatment and valorization strategies, as well as their overall environmental impact.

Reviewer 3 Report

The authors discuss a complete analysis of jarositum, which is a waste product burdening the environment. A complete analysis of waste jarorsite provides a comprehensive view of the composition and origin of toxic elements. Analyzes are comprehensive and their results are presented clearly. Possible outputs from the detected data could be better discussed, such as technological changes for the future, as well as the design of a possible technology to deal with already accumulated waste.

The quality of English is satisfactory, minor text corrections and editing could help the clarity of the text.
